# Impact of Adding GLP-1 Receptor Agonists to Insulin Therapy on Cardiovascular and Microvascular Outcomes in Type 2 Diabetes: A Nationwide Cohort Study from Taiwan

**DOI:** 10.3390/ph18091368

**Published:** 2025-09-12

**Authors:** Fu-Shun Yen, James Cheng-Chung Wei, Chen-Yu Sung, Pei-Yun Li, Fuu-Jen Tsai, Chih-Cheng Hsu, Chii-Min Hwu

**Affiliations:** 1Dr. Yen’s Clinic, No. 15, Shanying Road, Gueishan District, Taoyuan 33354, Taiwan; yenfushun@gmail.com; 2Department of Allergy, Immunology & Rheumatology, Chung Shan Medical University Hospital, No. 110, Sec. 1, Jianguo N. Rd., South District, Taichung 40201, Taiwan; jccwei@gmail.com; 3Institute of Medicine, Chung Shan Medical University, Taichung 40201, Taiwan; 4Graduate Institute of Integrated Medicine, China Medical University, Taichung 40402, Taiwan; 5Management Office for Health Data, China Medical University Hospital, Taichung 40447, Taiwanpeiyun.cmuh1999@gmail.com (P.-Y.L.); 6College of Medicine, China Medical University, Taichung 40402, Taiwan; 000704@tool.caaumed.org.tw; 7School of Chinese Medicine, College of Chinese Medicine, China Medical University, Taichung 40402, Taiwan; 8Department of Medical Research, China Medical University Hospital, Taichung 40402, Taiwan; 9Division of Medical Genetics, China Medical University Children’s Hospital, Taichung 40447, Taiwan; 10Institute of Population Health Sciences, National Health Research Institutes, 35 Keyan Road, Zhunan, Miaoli County 35053, Taiwan; 11Department of Health Services Administration, China Medical University, Taichung 40402, Taiwan; 12Department of Family Medicine, Min-Sheng General Hospital, 168 ChingKuo Road, Taoyuan 33044, Taiwan; 13National Center for Geriatrics and Welfare Research, National Health Research Institutes, No. 8, Xuefu West Road, Huwei Township, Yunlin County 632007, Taiwan; 14Faculty of Medicine, National Yang Ming Chiao Tung University School of Medicine, No. 155, Sec. 2, Linong Street, Taipei 11221, Taiwan; 15Section of Endocrinology and Metabolism, Department of Medicine, Taipei Veterans General Hospital, No. 201, Sec. 2, Shipai Road, Beitou District, Taipei 11217, Taiwan

**Keywords:** coronary artery disease, stroke, heart failure, end-stage kidney disease, sight-threatening retinopathy, amputation, all-cause mortality, cohort study, Taiwan, National Health Insurance Research Database

## Abstract

**Background/Objectives:** Selecting appropriate non-insulin hypoglycemic agents to complement insulin therapy is essential for achieving optimal glycemic control. This study aimed to evaluate the impact of glucagon-like peptide-1 receptor agonist (GLP-1 RA) plus insulin therapy on long-term cardiovascular and microvascular outcomes in patients with type 2 diabetes (T2D), with the goal of optimizing treatment strategies. **Methods:** Using Taiwan’s National Health Insurance Research Database (2008–2021), we conducted a retrospective cohort study and identified 6779 propensity score-matched pairs of insulin-treated patients with T2D who initiated either GLP-1 RAs or dipeptidyl peptidase-4 (DPP-4) inhibitors. Cox proportional hazard models were applied to compare outcome risks between the two groups. **Results:** The mean follow-up was 3.45 years. Compared with DPP-4 inhibitor use, GLP-1 RA use was significantly associated with a reduced risk of major adverse cardiovascular events (aHR 0.52, 95% CI 0.46–0.58), including hospitalizations for coronary artery disease (aHR 0.64, 95% CI 0.54–0.75), stroke (aHR 0.48, 95% CI 0.40–0.56), and heart failure (aHR 0.33, 95% CI 0.25–0.42). GLP-1 RA use was also linked to lower risks of major microvascular complications (aHR 0.42, 95% CI 0.35–0.50), end-stage kidney disease (aHR 0.08, 95% CI 0.04–0.14), sight-threatening retinopathy (aHR 0.62, 95% CI 0.50–0.76), leg amputation (aHR 0.16, 95% CI 0.05–0.57), and all-cause mortality (aHR 0.38, 95% CI 0.32–0.44). **Conclusions:** In this nationwide cohort, adding GLP-1 RAs to insulin therapy in patients with T2D was associated with significantly lower risks of cardiovascular events, major microvascular complications, and all-cause mortality compared with adding DPP-4 inhibitors. These findings suggest that incorporating GLP-1 RAs into insulin regimens may optimize treatment, lessen disease burden, and improve survival.

## 1. Introduction

At the time of diagnosis, individuals with type 2 diabetes (T2D) have typically lost approximately 50% of their β-cell secretory capacity, with subsequent annual declines ranging from about 2% in older adults to as high as 40% in younger patients or those with rapidly progressive disease [1,2,3,4,5]. Consequently, insulin therapy eventually becomes inevitable. Previous randomized trials have demonstrated that intensive insulin therapy significantly reduces the risk of both microvascular and macrovascular complications, underscoring the importance of early and rigorous glycemic control in improving long-term outcomes among individuals with T2D [6,7,8]. Nevertheless, nearly 70% of patients with T2D fail to achieve adequate glycemic control even while receiving insulin therapy [9]. Systematic reviews have shown that combination insulin therapy provides better glycemic control than insulin monotherapy [10,11]. However, even with basal insulin combined with oral hypoglycemic agents, approximately 40–50% of patients still fail to achieve adequate glycemic control after 24 weeks of treatment [12].

Selecting appropriate non-insulin hypoglycemic agents to complement insulin therapy is therefore crucial for optimizing both glycemic control and long-term outcomes [13]. Major classes of non-insulin agents include sulfonylureas, metformin, dipeptidyl peptidase-4 (DPP-4) inhibitors, sodium–glucose cotransporter-2 (SGLT-2) inhibitors, and glucagon-like peptide-1 receptor agonists (GLP-1 RAs), each with distinct mechanisms of action and clinical profiles [13,14]. While sulfonylureas may increase the risk of hypoglycemia and mortality when combined with insulin, metformin enhances insulin sensitivity, and DPP-4 inhibitors have shown protective effects against stroke [15,16]. GLP-1 RAs offer multiple advantages, including improved glycemic control, weight reduction, and a lower risk of hypoglycemia, though their use may be limited by gastrointestinal side effects, cost, and the requirement for injections [10,17,18]. Current international guidelines increasingly recommend GLP-1 RAs and SGLT-2 inhibitors for patients at elevated cardiovascular risk [19].

Despite these recognized benefits, evidence regarding the long-term cardiovascular and microvascular effects of adding GLP-1 RAs to insulin therapy remains limited [10,11]. To address this knowledge gap, we conducted a retrospective cohort study using Taiwan’s National Health Insurance Research Database. The primary objective was to evaluate the impact of adding GLP-1 RAs to insulin therapy on major cardiovascular and microvascular outcomes. A secondary objective was to compare the individual macrovascular and microvascular outcomes of adding GLP-1 RAs with those achieved by adding either DPP-4 inhibitors or sulfonylureas.

## 2. Results

### 2.1. Study Subjects

From the NHIRD, a total of 893,990 patients with T2D receiving insulin therapy were initially identified. After applying exclusion criteria and propensity score matching, 6779 matched pairs of GLP-1 RA users versus DPP-4 inhibitor users and 5242 matched pairs of GLP-1 RA users versus sulfonylurea users were obtained (Appendix A). Baseline demographics, comorbidities, concomitant medications, and diabetes duration were well balanced across groups, with all standardized mean differences (SMDs) < 0.1 (Table 1 and Appendix A). In the matched cohorts (GLP-1 RA vs. DPP-4 inhibitors and GLP-1 RA vs. sulfonylureas), the proportion of men was 53.32% and 51.50%, the mean (SD) age was 51.24 (12.67) and 52.66 (13.33) years, the mean (SD) diabetes duration was 7.17 (3.71) and 7.14 (3.76) years, and the mean (SD) follow-up was 3.45 (1.99) and 3.43 (2.00) years, respectively.

### 2.2. Key Findings

#### 2.2.1. GLP-1 RA Versus DPP-4 Inhibitor Use

In the propensity score-matched cohort comparing GLP-1 RAs with DPP-4 inhibitors, GLP-1 RA treatment was significantly associated with a lower risk of MACE [aHR 0.52, 95% CI 0.46–0.58; RR 0.54, 95% CI 0.49–0.61, corresponding to a 46% risk reduction], as well as hospitalizations for coronary artery disease (aHR 0.64, 95% CI 0.54–0.75), stroke (aHR 0.48, 95% CI 0.40–0.56), and heart failure (aHR 0.33, 95% CI 0.25–0.42). GLP-1 RA use was also associated with reduced risks of major microvascular complications [aHR 0.42, 95% CI 0.35–0.50; RR 0.39, 95% CI 0.33–0.47, a 61% reduction], end-stage kidney disease (aHR 0.08, 95% CI 0.04–0.14), sight-threatening retinopathy (aHR 0.62, 95% CI 0.50–0.76), leg amputation (aHR 0.16, 95% CI 0.05–0.57), and all-cause mortality (aHR 0.38, 95% CI 0.32–0.44) (Table 2). Kaplan–Meier analyses further showed significantly lower cumulative incidences of MACE, major microvascular complications, all-cause mortality, and hospitalizations for coronary artery disease, stroke, heart failure, end-stage kidney disease, and sight-threatening retinopathy in GLP-1 RA users compared with DPP-4 inhibitor users, with all log-rank tests exhibiting *p* < 0.001 (Figure 1, Appendix A). Sensitivity analysis excluding SGLT2 inhibitor users confirmed that the observed benefits of GLP-1 RAs were independent of concomitant SGLT2 inhibition (Appendix A).

#### 2.2.2. GLP-1 RA Versus Sulfonylurea Use

In the matched cohort comparing GLP-1 RAs with sulfonylureas, GLP-1 RA treatment was significantly associated with reduced risks of MACE (aHR 0.66, 95% CI 0.58–0.75), hospitalization for coronary artery disease (aHR 0.74, 95% CI 0.61–0.88), stroke (aHR 0.64, 95% CI 0.54–0.76), heart failure (aHR 0.54, 95% CI 0.42–0.70), major microvascular complications (aHR 0.68, 95% CI 0.56–0.82), end-stage kidney disease (aHR 0.39, 95% CI 0.27–0.57), leg amputation (aHR 0.29, 95% CI 0.09–0.91), and all-cause mortality (aHR 0.46, 95% CI 0.39–0.54). By contrast, the reduction in sight-threatening retinopathy risk did not reach statistical significance (aHR 0.85, 95% CI 0.67–1.08) (Table 3). Kaplan–Meier analyses showed significantly lower cumulative incidences of MACE (log-rank *p* < 0.001), major microvascular complications (log-rank *p* < 0.001), all-cause mortality (log-rank *p* < 0.001), hospitalization for coronary artery disease (log-rank *p* = 0.003), stroke (log-rank *p* < 0.001), heart failure (log-rank *p* < 0.001), and ESKD (log-rank *p* < 0.001) among GLP-1 RA users compared with sulfonylurea users (Figure 2, Appendix A).

### 2.3. Subgroup Analyses

#### 2.3.1. GLP-1 RA Versus DPP-4 Inhibitor Use

Across multiple subgroups, GLP-1 RA use was consistently associated with reduced risks of hospitalization for coronary artery disease, stroke, heart failure, end-stage kidney disease, sight-threatening retinopathy, and all-cause mortality compared with DPP-4 inhibitor use (Appendix A).

#### 2.3.2. GLP-1 RA Versus Sulfonylurea Use

In subgroup analyses, GLP-1 RA therapy was generally associated with lower risks of hospitalization for coronary artery disease, stroke, heart failure, ESKD, and all-cause mortality compared with sulfonylurea use (Appendix A).

## 3. Discussion

This nationwide cohort study in Taiwan demonstrated that, among patients with type 2 diabetes receiving insulin therapy, the addition of GLP-1 RAs was associated with reduced risks of MACE, hospitalizations for stroke, coronary artery disease, and heart failure, as well as major microvascular complications, end-stage kidney disease, sight-threatening retinopathy, leg amputation, and all-cause mortality, compared with the addition of DPP-4 inhibitors or sulfonylureas.

Long-term follow-up from the United Kingdom Prospective Diabetes Study (UKPDS) and the Diabetes Control and Complications Trial/Epidemiology of Diabetes Interventions and Complications (DCCT/EDIC) suggested that intensive glycemic control with insulin may reduce the risk of myocardial infarction [6,7]. In contrast, observational studies have indicated that insulin therapy may be associated with an increased risk of cardiovascular events and mortality [13,14,18]. A previous meta-analysis of randomized controlled trials of GLP-1 RAs demonstrated significant reductions in the risk of MACE and myocardial infarction in patients with T2D [20]. On this basis, the American Diabetes Association recommends combining GLP-1 RAs with insulin to improve treatment efficacy, sustain long-term benefits, and reduce cardiovascular risk [19]. Consistent with these recommendations, our study showed that in insulin-treated patients with T2D, GLP-1 RA use significantly reduced the risk of hospitalization for coronary artery disease compared with DPP-4 inhibitor and sulfonylurea use.

T2D is also associated with an increased risk, severity, and recurrence of stroke [21]. Meta-analyses of GLP-1 RAs have shown significantly lower risks of both total and non-fatal stroke compared with placebo [20,21]. Our previous study further indicated that DPP-4 inhibitor use in insulin-treated patients with T2D was linked to a lower stroke risk compared with non-DPP-4 inhibitor use [15]. In the present study, GLP-1 RA therapy was associated with a significantly reduced risk of hospitalization for stroke compared to both DPP-4 inhibitors and sulfonylureas in patients with T2D on insulin.

Compared with individuals without T2D, patients with T2D have a two- to four-fold higher prevalence of heart failure (HF) [22]. Among patients with both T2D and HF, approximately 30% are treated with insulin [23]. However, the weight gain and sodium retention induced by insulin may accelerate HF progression [14,17]. Therefore, selecting appropriate non-insulin adjunctive agents is critical in insulin-treated patients to mitigate worsening HF [10]. A meta-analysis reported that GLP-1 RA therapy significantly reduced the risk of HF hospitalization compared with placebo [20]. Similarly, our study demonstrated that GLP-1 RA use in insulin-treated patients was associated with a significantly lower risk of HF hospitalization than use of DPP-4 inhibitors or sulfonylureas. Collectively, the evidence suggests that GLP-1 RAs may help attenuate HF progression in insulin-treated patients with T2D.

Randomized controlled trials have shown that intensive insulin therapy reduces microvascular complications, particularly diabetic kidney disease [6,7,8]. However, insulin-treated patients generally have more severe and long-standing T2D, placing them at higher risk of kidney disease progression [22]. A meta-analysis has shown that GLP-1 RAs significantly reduce composite kidney outcomes, primarily by slowing progression to macroalbuminuria [19]. More recently, the FLOW trial demonstrated that subcutaneous semaglutide significantly reduced the risk of worsening renal function in patients with T2D and chronic kidney disease [24]. In line with these findings, our study showed that GLP-1 RA use in insulin-treated patients was associated with a lower risk of end-stage kidney disease compared to DPP-4 inhibitors or sulfonylureas.

Earlier studies suggested that insulin therapy may increase the risk of diabetic retinopathy, possibly due to rapid glucose lowering [25]. One randomized trial even reported a potential association between GLP-1 RA use and severe retinopathy [26]. By contrast, our previous cohort study found no significant increase in the risk of sight-threatening retinopathy with GLP-1 RA use compared with non-use, and in fact demonstrated a significantly lower risk compared with DPP-4 inhibitor use [27]. The present study further confirmed that, among insulin-treated patients with T2D, GLP-1 RA therapy was associated with a significantly lower risk of sight-threatening retinopathy compared with DPP-4 inhibitors.

Patients on insulin therapy often have poorer glycemic control and a higher burden of complications [14,17]. Amputation is one of the most disabling complications, leading to substantial disability and impaired quality of life [28]. Observational data suggest that GLP-1 RA use is linked to a significantly lower risk of amputation compared with DPP-4 inhibitor use [29]. Consistently, our study showed that among insulin-treated patients with T2D, GLP-1 RA therapy was associated with a markedly lower risk of amputation than either DPP-4 inhibitors or sulfonylureas.

Our previous research demonstrated that the combination of DPP-4 inhibitors with insulin was associated with reduced mortality risk in insulin-treated patients with T2D [30]. In addition, meta-analyses of cardiovascular outcome trials have shown that GLP-1 RA therapy significantly reduces cardiovascular death and all-cause mortality compared with placebo [19]. Furthermore, meta-analyses have indicated that combining GLP-1 RAs with insulin improves glycemic control, body weight, and blood pressure more effectively than insulin alone [10,11]. Together, these benefits may contribute to lowering both macrovascular and microvascular complications and reducing mortality. Our present study demonstrated that in insulin-treated patients with T2D, GLP-1 RA therapy was associated with a significantly lower risk of all-cause mortality compared with DPP-4 inhibitors and sulfonylureas.

The mechanisms underlying the protective effects of GLP-1 RAs against macrovascular and microvascular complications in T2D are likely multifactorial: (i) promotion of weight loss, reduction in systolic blood pressure, and improvement in lipid levels—key cardiovascular risk factors [17,20]; (ii) direct cardioprotective effects, including reduced ischemia-reperfusion injury, improved contractility, decreased fibrosis, and enhanced endothelial function, along with neuroprotection that may reduce risks of HF, myocardial infarction, and stroke [21,31]; (iii) lowering of blood glucose, oxidative stress, and inflammation, thereby slowing the progression of retinopathy, neuropathy, and atherosclerosis—drivers of both microvascular and macrovascular disease [11,31]; and (iv) renoprotection through reduced glomerular hyperfiltration and enhanced natriuresis, which ease renal stress and preserve kidney function [31].

### 3.1. Perspectives for Clinical Practice

Our findings indicate that, among patients with T2D receiving insulin therapy, the addition of GLP-1 RAs may confer substantial long-term benefits compared with DPP-4 inhibitors or sulfonylureas. GLP-1 RA use was associated with significant reductions in major cardiovascular events, critical microvascular complications (including kidney failure, sight-threatening retinopathy, and amputation), and all-cause mortality. These results underscore the role of GLP-1 RAs not only as effective glucose-lowering agents but also as therapies that improve prognosis and reduce overall disease burden.

In practice, GLP-1 RAs may be particularly advantageous for insulin-treated patients with elevated cardiovascular or microvascular risk. When clinically appropriate, they can be combined with metformin or SGLT2 inhibitors to further enhance cardiometabolic protection. Careful monitoring for adverse events is essential, and practical considerations such as treatment cost and accessibility should be taken into account to support equitable and effective implementation.

### 3.2. Limitations

This study has several limitations. First, the NHIRD lacks information on family history, physical activity, and dietary habits. Clinical data such as blood cholesterol, glucose levels, hemoglobin A1C, renal function, and inflammatory biomarkers are also unavailable. To mitigate these limitations, we used propensity score matching to balance important covariates between groups, including the DCSI, CCI, use of oral antidiabetic drugs, and diabetes duration, which may serve as proxies for disease severity. Second, medication adherence, including prescribed insulin doses, could not be reliably assessed from claims data. We also did not analyze hypoglycemia events, as outpatient hypoglycemia is often underreported in the database. Third, the study was conducted primarily in a Taiwanese population, which may limit generalizability to other ethnic groups. Finally, as with any cohort study, residual confounding and unmeasured biases cannot be excluded, and causal inference remains limited. Further confirmation from randomized controlled trials is warranted.

## 4. Materials and Methods

### 4.1. Study Population and Data Source

This retrospective cohort study used data from Taiwan’s National Health Insurance Research Database (NHIRD), which covers nearly 99% of the national population. The database contains comprehensive healthcare information, including demographics, clinical diagnoses, medical procedures, and prescription records [32]. Diagnoses were coded using the International Classification of Diseases, Ninth and Tenth Revisions, Clinical Modification (ICD-9-CM and ICD-10-CM). Mortality data were validated through linkage with the National Death Registry to improve accuracy. Ethical approval was obtained from the Research Ethics Committee of China Medical University Hospital (CMUH110-REC1-038 [CR-3]). As all data were de-identified before analysis, the need for informed consent was waived.

This study was designed and reported in accordance with the STROBE guidelines and checklist [33].

### 4.2. Study Design and Procedures

Patients with T2D were identified in the NHIRD between 1 January 2008, and 31 December 2021 (Appendix A). T2D was defined as having at least two outpatient visits or one hospitalization with a T2D diagnosis within a single year, based on outpatient and/or inpatient records (Appendix A) [34]. The validity of using ICD codes for identifying T2D in Taiwan has been confirmed, with an accuracy of 93.3% [34]. From this cohort, patients with newly diagnosed T2D who initiated insulin therapy were included.

Exclusion criteria were as follows: (1) age < 20 or >100 years; (2) missing data on age or sex; (3) history of type 1 diabetes, dialysis, sight-threatening retinopathy (defined as ≥2 outpatient visits or ≥1 hospitalization with surgical intervention, laser photocoagulation within 90 days of diagnosis, vision loss, or anti-VEGF injection), or leg amputation before the index date; (4) use of GLP-1 RAs or DPP-4 inhibitors within 3 months before insulin initiation; (5) concurrent use of GLP-1 RAs and DPP-4 inhibitors; and (6) not receiving either of these agents.

Patients who initiated GLP-1 RAs after their T2D diagnosis were classified as GLP-1 RA users, while those who began DPP-4 inhibitors or sulfonylureas were categorized accordingly. The index date for each patient was defined as the date of initiating the respective medication. Since GLP-1 RAs and DPP-4 inhibitors became available in Taiwan only after 2011, the index date for all groups was set on or after 1 January 2011. Patients were followed from the index date until 31 December 2021.

### 4.3. Demographics and Related Variables of the Participants

We adjusted for a range of clinically relevant variables that could influence the likelihood of receiving GLP-1 RA therapy and thereby affect outcomes (Table 1 and Appendix A). These included demographic factors (age, sex), as well as comorbidities present before the index date, such as overweight/obesity, smoking, alcohol-related disorders, hypertension, dyslipidemia, coronary artery disease, stroke, heart failure, atrial fibrillation, peripheral arterial disease, chronic obstructive pulmonary disease, liver cirrhosis, chronic kidney disease, retinopathy, and cancer.

Medication use was also considered, including the number and classes of oral antidiabetic agents, antihypertensives, aspirin, and statins. In addition, we included the Charlson Comorbidity Index (CCI) [35], the Diabetes Complications Severity Index (DCSI) [36], and diabetes duration.

### 4.4. Main Outcomes

The primary outcomes were: (1) major adverse cardiovascular events (MACEs), defined as a composite of hospitalizations for coronary artery disease (CAD), stroke, and heart failure, and (2) major microvascular complications, defined as a composite of end-stage kidney disease (ESKD), sight-threatening retinopathy, and leg amputation [34]. Secondary outcomes included each of these components individually—hospitalizations for CAD, stroke, heart failure, ESKD, sight-threatening retinopathy, non-traumatic lower limb amputation—as well as all-cause mortality. Mortality and causes of death were confirmed using death certificates and data from the National Death Registry. Patients were followed until the occurrence of any outcome, death, or 31 December 2021, whichever came first.

### 4.5. Statistical Analysis

To ensure comparability between GLP-1 RA users and non-users, we performed 1:1 propensity score matching. Propensity scores were estimated using a non-parsimonious multivariable logistic regression model, with GLP-1 RA initiation as the dependent variable. Covariates included demographic characteristics (age, sex), smoking status, obesity, comorbidities, CCI and DCSI scores (≤1 vs. >1), index year, medication use, and duration of T2D (Table 1 and Appendix A). Differences between groups were evaluated using Student’s *t*-tests for continuous variables and chi-squared tests for categorical variables. Matching was conducted using the nearest-neighbor method, with balance considered acceptable when the standardized mean difference (SMD) was <0.1 or the *p*-value > 0.05.

Outcomes were compared between GLP-1 RA users and DPP-4 inhibitor/sulfonylurea users using Cox proportional hazards models with robust sandwich variance estimators. The proportional hazards assumption was assessed using Schoenfeld residuals. Results are expressed as hazard ratios (HRs) with 95% confidence intervals (CIs). Cumulative incidence of outcomes was estimated using Kaplan–Meier curves and compared with log-rank tests.

Subgroup analyses were performed to assess effect modification by sex, age, hypertension, dyslipidemia, CCI, DCSI, number of oral antidiabetic agents, statin use, and diabetes duration. A sensitivity analysis was conducted excluding SGLT2 inhibitor users from the GLP-1 RA and DPP-4 inhibitor cohorts to confirm that the observed benefits of GLP-1 RAs were not attributable to concomitant SGLT2 inhibition.

A two-sided *p*-value < 0.05 was considered statistically significant. All analyses were performed using SAS software (version 9.4; SAS Institute, Cary, NC, USA).

## 5. Conclusions

In this nationwide cohort of patients with type 2 diabetes receiving insulin therapy, the addition of GLP-1 RAs was associated with significantly lower risks of cardiovascular events, major microvascular complications, and all-cause mortality compared with the addition of DPP-4 inhibitors or sulfonylureas. These findings suggest that incorporating GLP-1 RAs into insulin regimens may help optimize treatment, reduce disease burden, and improve survival.

Nevertheless, as this was an observational study, the possibility of residual confounding must be acknowledged, and the generalizability of our findings beyond the Taiwanese healthcare setting may be limited. Further randomized controlled trials and investigations in more diverse populations are warranted to confirm these benefits and inform clinical practice.

## Figures and Tables

**Figure 1 pharmaceuticals-18-01368-f001:**
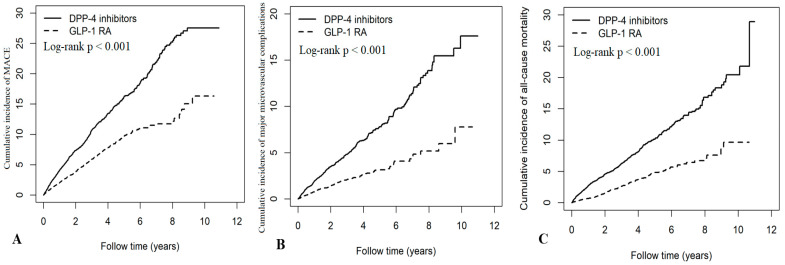
Cumulative incidence curves for (**A**) major adverse cardiovascular events (MACEs), (**B**) major microvascular complications, and (**C**) all-cause mortality among GLP-1 RA users versus DPP-4 inhibitor users.

**Figure 2 pharmaceuticals-18-01368-f002:**
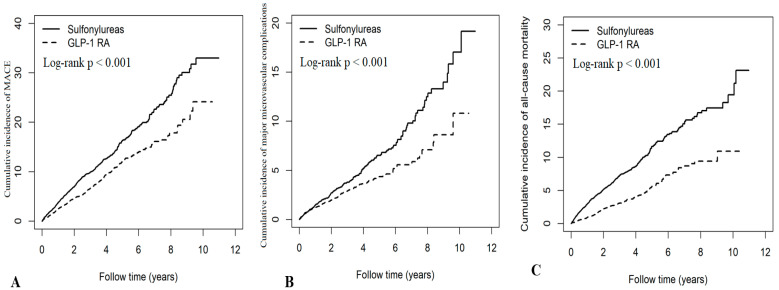
Cumulative incidence curves for (**A**) major adverse cardiovascular events (MACEs), (**B**) major microvascular complications, and (**C**) all-cause mortality among GLP-1 RA users versus sulfonylurea users.

**Table 1 pharmaceuticals-18-01368-t001:** Characteristics of patients with T2D treated with insulin and DPP-4 inhibitor or GLP-1 RA.

Variable	DPP-4 Inhibitor Users		GLP-1 RA Users		SMD *	*p*-Value
	(N = 6779)		(N = 6779)			
	n	%	n	%		
Gender					0.007	0.705
Female	3175	46.84	3153	46.51		
Male	3604	53.16	3626	53.49		
Age, years						0.202
20–40	1479	21.82	1528	22.54	0.017	
41–60	3600	53.11	3549	52.35	0.015	
61–80	1656	24.43	1639	24.18	0.006	
81–100	44	0.65	63	0.93	0.032	
Mean ± SD	51.27	12.48	51.21	12.87	0.004	0.804
Comorbidities						
Obesity	721	10.64	774	11.42	0.025	0.146
Smoking	437	6.45	407	6.00	0.018	0.286
Alcohol-related disorders	216	3.19	209	3.08	0.006	0.73
Hypertension	4475	66.01	4489	66.22	0.004	0.799
Dyslipidemia	5769	85.10	5727	84.48	0.017	0.315
Coronary artery disease	952	14.04	980	14.46	0.012	0.492
Stroke	401	5.92	408	6.02	0.004	0.8
Heart failure	110	1.62	133	1.96	0.026	0.137
Atrial fibrillation	425	6.27	436	6.43	0.007	0.698
Peripheral artery disease	238	3.51	232	3.42	0.005	0.778
COPD	564	8.32	582	8.59	0.010	0.578
Cirrhosis	134	1.98	120	1.77	0.015	0.375
Chronic kidney disease	721	10.64	736	10.86	0.007	0.677
Retinopathy	1701	25.09	1677	24.74	0.008	0.634
Cancers	269	3.97	288	4.25	0.014	0.411
Charlson Comorbidity Index					0.011	0.516
≤1	5464	80.60	5434	80.16		
>1	1315	19.40	1345	19.84		
Diabetes Complications Severity Index					0.011	0.535
≤1	3708	54.70	3672	54.17		
>1	3071	45.30	3107	45.83		
Medications						
Sulfonylurea	5591	82.48	5609	82.74	0.007	0.683
SGLT2 inhibitor	1994	29.41	2236	32.98	0.077	<0.001
Thiazolidinedione	3081	45.45	3062	45.17	0.006	0.743
Alpha-glucosidase inhibitor	2732	40.30	2798	41.27	0.020	0.249
Metformin	6544	96.53	6555	96.70	0.009	0.601
Premix insulin	2082	30.71	2022	29.83	0.019	0.262
Short-acting insulin	3177	46.87	3192	47.09	0.004	0.796
Basal insulin	5771	85.13	5775	85.19	0.002	0.923
ACEI	2005	29.58	2002	29.53	0.001	0.955
ARB	4010	59.15	3971	58.58	0.012	0.496
α-blocker	597	8.81	578	8.53	0.010	0.562
β-blocker	3285	48.46	3299	48.66	0.004	0.81
Calcium-channel blocker	3735	55.10	3770	55.61	0.010	0.545
Diuretic	2480	36.58	2446	36.08	0.010	0.544
Statin	5212	76.88	5134	75.73	0.027	0.115
Aspirin	2892	42.66	2905	42.85	0.004	0.821
Number of oral antidiabetic drugs					0.004	0.838
0–3	1539	22.70	1549	22.85		
>3	5240	77.30	5230	77.15		
Duration of T2D, years					0.005	0.778
≤5	2035	30.02	2020	29.80		
>5	4744	69.98	4759	70.20		
Mean ± SD	7.20	3.77	7.13	3.65	0.019	0.271

* A standardized mean difference (SMD) below 0.1 or a *p*-value greater than 0.05 was regarded as indicating negligible differences between GLP-1 RA and DPP-4 inhibitor users.

**Table 2 pharmaceuticals-18-01368-t002:** The outcome risks of adding DPP-4 inhibitors versus GLP-1 RA to insulin therapy in patients with T2D.

Outcome	DPP-4 Inhibitor Users			GLP-1 RA Users			cHR	(95% CI)	aHR ^†^	(95% CI)
	n	PY	IR	n	PY	IR				
Primary outcomes										
Major adverse cardiovascular events ^a^	777	21,421	36.27	467	23,647	19.75	0.54	(0.48, 0.61) ***	0.52	(0.46, 0.58) ***
Major microvascular outcomes ^b^	385	22,248	17.30	165	24,283	6.79	0.40	(0.33, 0.48) ***	0.42	(0.35, 0.50) ***
Secondary outcomes										
Hospitalization for coronary artery disease	359	22,317	16.09	258	24,087	10.71	0.66	(0.56, 0.78) ***	0.64	(0.54, 0.75) ***
Hospitalization for stroke	400	22,311	17.93	212	24,221	8.75	0.49	(0.41, 0.58) ***	0.48	(0.40, 0.56) ***
Hospitalization for heart failure	231	22,656	10.20	81	24,508	3.31	0.33	(0.25, 0.42) ***	0.33	(0.25, 0.42) ***
End-stage kidney disease	158	22,826	6.92	12	24,635	0.49	0.07	(0.04, 0.13) ***	0.08	(0.04, 0.14) ***
Sight-threatening retinopathy	234	22,538	10.38	150	24,308	6.17	0.60	(0.49, 0.73) ***	0.62	(0.50, 0.76) ***
Leg amputation	20	23,114	0.87	3	24,641	0.12	0.15	(0.05, 0.52) **	0.16	(0.05, 0.57) **
All-cause mortality	519	23,136	22.43	225	24,650	9.13	0.41	(0.35, 0.48) ***	0.38	(0.32, 0.44) ***

^†^ aHR adjusted for age, gender, comorbidities, CCI, DCSI, concomitant medications, and duration of T2D, as detailed in Table 1. ^a^ Composite outcome includes hospitalizations for coronary artery disease, stroke, and heart failure. ^b^ Composite outcome includes end-stage kidney disease, sight-threatening retinopathy, and non-traumatic leg amputation. ** *p* < 0.01, *** *p* < 0.001.

**Table 3 pharmaceuticals-18-01368-t003:** The outcome risks of adding sulfonylurea versus GLP-1 RA to insulin therapy in patients with T2D.

Outcome	Sulfonylurea Users			GLP-1 RA Users			cHR	(95% CI)	aHR ^†^	(95% CI)
	n	PY	IR	n	PY	IR				
Primary outcomes										
Major adverse cardiovascular events ^a^	595	16,482	36.10	441	18,184	24.25	0.68	(0.60, 0.77) ***	0.66	(0.58, 0.75) ***
Major microvascular outcomes ^b^	242	17,124	14.13	177	18,631	9.50	0.69	(0.57, 0.83) ***	0.68	(0.56, 0.82) ***
Secondary outcomes										
Hospitalization for coronary artery disease	264	17,101	15.44	216	18,630	11.59	0.76	(0.64, 0.91) **	0.74	(0.61, 0.88) **
Hospitalization for stroke	325	17,060	19.05	225	18,598	12.10	0.64	(0.54, 0.76) ***	0.64	(0.54, 0.76) ***
Hospitalization for heart failure	160	17,399	9.20	92	18,904	4.87	0.54	(0.41, 0.69) ***	0.54	(0.42, 0.70) ***
End-stage kidney disease	100	17,478	5.72	39	18,988	2.05	0.37	(0.26, 0.54) ***	0.39	(0.27, 0.57) ***
Sight-threatening retinopathy	148	17,332	8.54	137	18,696	7.33	0.87	(0.69, 1.09)	0.85	(0.67, 1.08)
Leg amputation	16	17,664	0.91	4	19,045	0.21	0.26	(0.09, 0.79) *	0.29	(0.09, 0.91) *
All-cause mortality	431	17,695	24.36	210	19,052	11.02	0.46	(0.39, 0.54) ***	0.46	(0.39, 0.54) ***

^†^ aHR values were adjusted for age, sex, comorbidities, CCI, DCSI, medication use, and duration of T2D (see Appendix A). ^a^ Composite outcome includes hospitalizations for coronary artery disease, stroke, and heart failure. ^b^ Composite outcome includes end-stage kidney disease, sight-threatening retinopathy, and lower-limb amputation. * *p* < 0.05, ** *p* < 0.01, *** *p* < 0.001.

## Data Availability

Data for this study derive from Taiwan’s National Health Insurance Research Database (NHIRD), maintained by the National Health Insurance Administration. In accordance with Taiwan’s Personal Information Protection Act (effective 2012), the datasets cannot be shared in the article, Appendix A, or a public repository. Investigators may request access by submitting a formal application to the NHIRD Office or by emailing stsung@mohw.gov.tw.

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
