# Peer review of "Impact of Adding GLP-1 Receptor Agonists to Insulin Therapy on Cardiovascular and Microvascular Outcomes in Type 2 Diabetes: A Nationwide Cohort Study from Taiwan"

_pharmaceuticals, 2025, doi:10.3390/ph18091368_

Round 1

Reviewer 1 Report

Comments and Suggestions for Authors

Dear authors,

Please see my comments below:

  • Title should indicate the type of study and region
  • Abstract:
    • Please indicate if this study is prospective or retrospective… this formulation could lead to confusion “ From January 1, 2008, to December 31, 2021 …”
  • Introduction:
    • You should present the main classes of the non-insulin hypoglycemic agents. Also, a short description of their mechanism of action and advantages would be useful.
    • What do current guidelines recommend for selecting antidiabetic therapies, especially GLP-1 receptor agonists?
    • What are the disadvantages of GLP-1?
    • The aim of the study should be improved. Please mention that it is a cohort study performed by comparison with iDDP-4.

  • Materials and methods:
    • Section 2.1: Are both hospitalizations and outpatient encounters captured in the dataset? Also, please indicate the type of study
    • Line 112: “We recruited patients …” – please check if this information is correct. Did you recruit patients??
    • Lines 130: “Patients not requiring GLP-1 RA or DPP-4 inhibitors” … How was established the clinical necessity of GLP-1 RAs or iDPP-4?
    • Please specify how you calculated CCI and DCSI? Were they treated as continuous or categorical variables in the propensity score model and subgroup analyses
  • Results:
    • All abbreviations should be explained in the tables/figure caption.
  • Discussions
    • Line 265: please mention the region
    • Line 344-356: another system of enumeration would be more useful (e.g. (i), (ii),…)
    • Line 362: please remove the supplementary “, ”
    • Lines 367-368: Probably “may not be generalizable to other ethnic groups” would be more appropriate than “may not be applicable…”

Author Response

Responses to the comments of Reviewer #1

Please see my comments below:

Title should indicate the type of study and region

Response: We are deeply grateful for your careful review of our manuscript and for your encouraging feedback. As reflected in the title, both the study type and the study region are clearly indicated: ‘Impact of Adding GLP-1 Receptor Agonists to Insulin Therapy on Cardiovascular and Microvascular Outcomes in Type 2 Diabetes: A Nationwide Cohort Study from Taiwan.’ We sincerely thank you once again for your thoughtful comments.

Abstract:

Please indicate if this study is prospective or retrospective… this formulation could lead to confusion “ From January 1, 2008, to December 31, 2021 …”

Response: Thank you for your suggestion. We have revised the sentence to: ‘Using Taiwan’s National Health Insurance Research Database (2008–2021), we conducted a retrospective cohort study and identified 6,779 propensity score–matched pairs of insulin-treated patients with type 2 diabetes who initiated either GLP-1 receptor agonists or dipeptidyl peptidase-4 (DPP-4) inhibitors.’ This revision clarifies that the study design was a retrospective cohort study.

Introduction:

You should present the main classes of the non-insulin hypoglycemic agents. Also, a short description of their mechanism of action and advantages would be useful.

What do current guidelines recommend for selecting antidiabetic therapies, especially GLP-1 receptor agonists?

What are the disadvantages of GLP-1?

The aim of the study should be improved. Please mention that it is a cohort study performed by comparison with iDDP-4.

Response: In accordance with your valuable suggestion, we have revised this section to present the main classes of non-insulin hypoglycemic agents, provide a brief description of their mechanisms of action and advantages, summarize current guideline recommendations for selecting antidiabetic therapies—particularly GLP-1 RAs—acknowledge the disadvantages of GLP-1 RAs, and clarify that this is a cohort study conducted in comparison with DPP-4 inhibitors (page 2).

Materials and methods:

Section 2.1: Are both hospitalizations and outpatient encounters captured in the dataset? Also, please indicate the type of study

Line 112: “We recruited patients …” – please check if this information is correct. Did you recruit patients??

Lines 130: “Patients not requiring GLP-1 RA or DPP-4 inhibitors” … How was established the clinical necessity of GLP-1 RAs or iDPP-4?

Please specify how you calculated CCI and DCSI? Were they treated as continuous or categorical variables in the propensity score model and subgroup analyses

Response: We are sincerely grateful for your thoughtful suggestion, and we have made the recommended revisions to all of the above items (pages 10-11).

Results:

All abbreviations should be explained in the tables/figure caption.

Response: All abbreviations were explained in the tables and figure captions (pages 4-7).

Discussions

Line 265: please mention the region

Line 344-356: another system of enumeration would be more useful (e.g. (i), (ii),…)

Line 362: please remove the supplementary “, ”

Lines 367-368: Probably “may not be generalizable to other ethnic groups” would be more appropriate than “may not be applicable…”

Response: We have carefully addressed all the points you raised in your comments and revised the manuscript accordingly. We sincerely thank you for your constructive suggestions, which have helped us improve the quality of our work (pages 7,9,10).

Reviewer 2 Report

Comments and Suggestions for Authors

Yen and colleagues examine the effects of GLP-1 receptor agonists (GLP-1 RAs)  on vascular complications in insulin-treated type 2 diabetes patients. While the manuscript provides substantial data, it would benefit from additional clinical context to strengthen the interpretation of findings.

Major points for consideration:

1. The introduction’s reference to a 4% decline in insulin secretion is overly general. Depending on cohort characteristics and age, this may range from 2% to 40%. Among older patients, the decline is typically around 2–5%.
2.  A brief discussion of DPP-4 inhibitors and sulfonylureas would enhance context in the introduction.
3. Tables should include p-values alongside standardized mean differences (SMDs) to aid interpretation.
4. Incorporating statistical analyses of key clinical parameters—lipid profile, glycemic measures, renal function, and inflammatory markers—would strengthen the manuscript.
5. The rationale for stratifying groups at age 60 is unclear. Younger patients often exhibit more rapid disease progression than those diagnosed after 40; cardiovascular guidelines frequently reference 40 years as a risk threshold. A more clinically justified stratification is recommended.
6.Given evidence that GLP-1 RAs reduce cardiovascular events in patients with comorbidity of type 2 diabetes and gout , this could be briefly addressed in the discussion.
7. In addition to hazard ratios, relative risk analysis might provide complementary insight into the protective effect of therapy, despite losing the time dimension.
8. All abbreviations should be listed in abbreviation section and defined at first mention in the text.

Author Response

Responses to the comments of Reviewer #2

Yen and colleagues examine the effects of GLP-1 receptor agonists (GLP-1 RAs)  on vascular complications in insulin-treated type 2 diabetes patients. While the manuscript provides substantial data, it would benefit from additional clinical context to strengthen the interpretation of findings.

Response: We sincerely appreciate your careful review of our manuscript and are grateful for your kind and encouraging comments.

Major points for consideration:

  1. The introduction’s reference to a 4% decline in insulin secretion is overly general. Depending on cohort characteristics and age, this may range from 2% to 40%. Among older patients, the decline is typically around 2–5%.

Response: Thank you for your valuable suggestion. We have revised the text accordingly to read: “At diagnosis, individuals with type 2 diabetes (T2D) have typically lost ~50% of their β-cell secretory capacity, with subsequent annual declines ranging from ~2% in older adults to as high as ~40% in younger or more rapidly progressive cases [1–5].”

  1. A brief discussion of DPP-4 inhibitors and sulfonylureas would enhance context in the introduction.

Response: Thank you for your suggestion. We have added a brief discussion of DPP-4 inhibitors and sulfonylureas in the Introduction to provide additional context (page 2).

  1. Tables should include p-values alongside standardized mean differences (SMDs) to aid interpretation.

Response: We have included the p-values alongside the standardized mean differences (SMDs) to facilitate interpretation (pages 3,4,11).

  1. Incorporating statistical analyses of key clinical parameters—lipid profile, glycemic measures, renal function, and inflammatory markers—would strengthen the manuscript.

Response: We regret that our National Health Insurance Research Database (NHIRD) does not contain data on lipid profiles, glycemic measures, renal function, or inflammatory biomarkers (page 9). As a result, we were unable to incorporate statistical analyses of these key clinical parameters to further strengthen the manuscript. This limitation has been explicitly addressed in the Discussion section. Thank you for your recommendation.

  1. The rationale for stratifying groups at age 60 is unclear. Younger patients often exhibit more rapid disease progression than those diagnosed after 40; cardiovascular guidelines frequently reference 40 years as a risk threshold. A more clinically justified stratification is recommended.

Response: Thank you for your suggestion. We have revised the age stratification for the matching analysis to the following categories: 20–40, 41–60, 61–80, and 81–100 years (page 3).

6.Given evidence that GLP-1 RAs reduce cardiovascular events in patients with comorbidity of type 2 diabetes and gout , this could be briefly addressed in the discussion.

Response: Thank you for your valuable suggestion. Our study did not specifically include a subgroup of patients with both type 2 diabetes and gout; therefore, we are unable to provide a targeted discussion on the cardiovascular effects of GLP-1 RAs in this population. We hope for your understanding.

  1. In addition to hazard ratios, relative risk analysis might provide complementary insight into the protective effect of therapy, despite losing the time dimension.

Response: In addition to reporting hazard ratios, we have included a relative risk analysis to provide complementary insights into the protective effects of the therapy (page 4). Thank you for your suggestion.

  1. All abbreviations should be listed in abbreviation section and defined at first mention in the text.

Response: We have ensured that all abbreviations are listed in the abbreviation section and defined at their first mention in the text (page 13). Thank you.

Reviewer 3 Report

Comments and Suggestions for Authors

Dear Authors,

I would like to express my sincere gratitude for the opportunity to contribute my opinion to the evaluation of your manuscript. I found the topic interesting and relevant to our field. Below, I list the main areas that could benefit from further elaboration and revision:

  • Editing: the elevated number of acronim require attention: I suggest a general check. Im'm not sure that the refernces are update according the journal template. Please adopt "gender" and not "sex" in the the full text. I suggest only one figure of summary of the study; please summary;
  • Title: I suggest to insert the type of study conducted (not full clear in other part of the manuscript) and the setting (e.g. Germany); this last element is fundamental for discussion of limitations (see under);
  • Abstract: an expansion of this section is needed in terms of “Clinical Practice view” (see under);
  • Keywords: I suggest 4/5 that relevant for principal topics studies, population and type of study conduct;
  • Introduction: full missing in epidemiologica data in international and national view. In complex the introduction of the manuscript is well-structured and provides a solid synthesis of the evidence regarding GLP-1 receptor agonists. However, since the study includes a comparison with DPP-4 inhibitors, it would be appropriate to further elaborate on the theoretical framework concerning this pharmacological class. An integrative analysis could, for example, include the recently discussed hypotheses on the potential protective role of DPP-4 inhibitors in patients with type 2 diabetes, with particular attention not only to cardiovascular and microvascular aspects, but also to possible extra-metabolic effects observed in complex settings such as emerging infectious diseases. Expanding this section would provide the reader with a more balanced overview and allow for a better understanding of the differences and potential synergies between the two drug classes, placing the study’s findings within a broader and more up-to-date perspective. For these reason I suggest to extend the section with the topis as "Potential role of incretins in diabetes and COVID-19 infection" and Efficacy and Safety of Sitagliptin in the Treatment of COVID-19" that certly linked relevant number of readers and researchers interest of data finding. Full answer of this element is fundamental for international consideration. In this section rest conflit in therm of objectives. For this reason, I recommend adopting the classic structure: “The primary objectives of the study were… while the secondary ones…” and eventually the research questions;
  • Methods: This section, in my view, required more attention. As above comment, not full clear in the text the study conducted and several elements are either missing or insufficiently described. The reporting tool is absent, as is the related checklist in the supplementary files and the corresponding reference depending on the study design (see: https://www.equator-network.org/). Using such a tool would help the authors to present this crucial section more clearly and comprehensively. This element, as the previous for the introduction, is fundamental for international consideration;
  • Results: Overall, this section is well done and arguably the strength of the study. It will certainly benefit from the previous and following suggestions. Please improve the vision of the flow of the study as previous recommended;
  • Discussion: I would suggest focusing more on clinical practice implications and “Perspectives for Clinical Practice” section could be add and suppose to extend the management of care of this patients in several point of view. You could also summarize the collected data in this light previous suggestion of the introduction and consider to update the study in support of data finding (see the comment for references);
  • Limitations: In my opinion, these should be addressed in the possible generalization of the collected data and specific section is recommended;
  • Conclusions: I recommend developing a critical analysis here as well, following the considerations previously outlined.
  • References: see the comments above and update the reference (over 10 years if not for method section or relevant evidence based data finding) and consider to update the references for DPP-4 and GLP-1 use in real world study that today are certly more update;
  • In summary, I suggest responding point by point to each individual suggestion for possible reconsideration, mainly for two relevant elements previous inidcated.
Comments on the Quality of English Language

Native english review suggested

Author Response

Responses to the comments of Reviewer #3

I would like to express my sincere gratitude for the opportunity to contribute my opinion to the evaluation of your manuscript. I found the topic interesting and relevant to our field. Below, I list the main areas that could benefit from further elaboration and revision:

Response: Thank you for reviewing our manuscript and for your encouraging comments.

Editing: the elevated number of acronyms require attention: I suggest a general check. I'm not sure that the references are update according to the journal template. Please adopt "gender" and not "sex" in the full text. I suggest only one figure of summary of the study; please summary.

Response: We conducted a general review to reduce the number of acronyms used in the manuscript, updated the references in accordance with the journal’s formatting requirements, replaced the term “sex” with “gender” throughout the text, and prepared a graphical abstract to summarize the study (page 3, 5,7,11, 13). 

Title: I suggest to insert the type of study conducted (not full clear in other part of the manuscript) and the setting (e.g. Germany); this last element is fundamental for discussion of limitations (see under);

Response: We revised the title to explicitly include the study design and setting, as follows: “Impact of Adding GLP-1 Receptor Agonists to Insulin Therapy on Cardiovascular and Microvascular Outcomes in Type 2 Diabetes: A Nationwide Cohort Study from Taiwan.”

Abstract: an expansion of this section is needed in terms of “Clinical Practice view” (see under);

Response: We have added an abstract section framed under “Clinical Practice View” to provide practical implications of our findings (page 2).

Keywords: I suggest 4/5 that relevant for principal topics studies, population and type of study conduct;

Response: We have included some relevant keywords that reflect the main topics, study population, and study design (page 2).

Introduction: full missing in epidemiological data in international and national view. In complex the introduction of the manuscript is well-structured and provides a solid synthesis of the evidence regarding GLP-1 receptor agonists. However, since the study includes a comparison with DPP-4 inhibitors, it would be appropriate to further elaborate on the theoretical framework concerning this pharmacological class. An integrative analysis could, for example, include the recently discussed hypotheses on the potential protective role of DPP-4 inhibitors in patients with type 2 diabetes, with particular attention not only to cardiovascular and microvascular aspects, but also to possible extra-metabolic effects observed in complex settings such as emerging infectious diseases. Expanding this section would provide the reader with a more balanced overview and allow for a better understanding of the differences and potential synergies between the two drug classes, placing the study’s findings within a broader and more up-to-date perspective. For these reason I suggest to extend the section with the topic as "Potential role of incretins in diabetes and COVID-19 infection" and Efficacy and Safety of Sitagliptin in the Treatment of COVID-19" that certly linked relevant number of readers and researchers interest of data finding. Full answer of this element is fundamental for international consideration. In this section rest conflit in therm of objectives. For this reason, I recommend adopting the classic structure: “The primary objectives of the study were… while the secondary ones…” and eventually the research questions.

Response: Thank you for your valuable suggestion. Following your advice, we have revised the objectives of this study by adopting the classic structure (page 2): The primary objective of this cohort study was to evaluate the impact of adding GLP-1 RAs to insulin therapy on major cardiovascular and microvascular outcomes. The secondary objective was to compare the individual macrovascular and microvascular outcomes with those achieved by adding either DPP-4 inhibitors or sulfonylureas.”

We sincerely appreciate your thoughtful suggestion to extend the section with the topic ‘Potential role of incretins in diabetes and COVID-19 infection’. However, our study was specifically designed to evaluate macrovascular and microvascular complications, and we had already prespecified nine outcomes. Including additional outcomes at this stage could raise concerns regarding multiple comparisons. Furthermore, all study outcomes must be determined a priori, as the selection of matching and adjustment variables is inherently dependent on their potential association with the predefined outcomes. In order to appropriately evaluate COVID-19 infection as an outcome, we would have needed to consider, at the study design stage, additional variables such as patients’ immune status, history of COVID-19 vaccination (including type and number of doses), and any prior COVID-19 infection. Since such variables were not incorporated into the original study design, we were unable to examine this outcome. We kindly ask for your understanding of these considerations.

Methods: This section, in my view, required more attention. As above comment, not full clear in the text the study conducted and several elements are either missing or insufficiently described. The reporting tool is absent, as is the related checklist in the supplementary files and the corresponding reference depending on the study design (see: https://www.equator-network.org/). Using such a tool would help the authors to present this crucial section more clearly and comprehensively. This element, as the previous for the introduction, is fundamental for international consideration.

Response: We are grateful for your thoughtful reminder. In response, we have added the following statement: ‘This study was conducted and reported in accordance with the STROBE guidelines and the accompanying checklist (page 10).

Results: Overall, this section is well done and arguably the strength of the study. It will certainly benefit from the previous and following suggestions. Please improve the vision of the flow of the study as previous recommended.

Response: Thank you for your encouraging comment. We have improved the clarity and flow of the Results section (pages 3-7).

Discussion: I would suggest focusing more on clinical practice implications and “Perspectives for Clinical Practice” section could be add and suppose to extend the management of care of this patients in several point of view. You could also summarize the collected data in this light previous suggestion of the introduction and consider to update the study in support of data finding (see the comment for references);

Response: We are grateful for your valuable suggestion. In response, we have added a section entitled Perspectives for Clinical Practice in the Discussion (page 9).

Limitations: In my opinion, these should be addressed in the possible generalization of the collected data and specific section is recommended.

Response: We have added a statement on the potential generalizability of the collected data in the Limitations subsection of the Discussion (page 10).

Conclusions: I recommend developing a critical analysis here as well, following the considerations previously outlined.

Response: We have developed a critical analysis in the Conclusion section, based on the considerations outlined above (pages 11-12).

References: see the comments above and update the reference (over 10 years if not for method section or relevant evidence based data finding) and consider to update the references for DPP-4 and GLP-1 use in real world study that today are certly more update;

Response: We sincerely appreciate your valuable suggestion. In response, we have updated the references concerning the real-world use of DPP-4 inhibitors and GLP-1 RAs (pages 13-15).

In summary, I suggest responding point by point to each individual suggestion for possible reconsideration, mainly for two relevant elements previous indicated.

Response: We are grateful for your thoughtful recommendations. We have carefully addressed each suggestion in a point-by-point manner, with particular emphasis on the two key elements highlighted above.

Comments on the Quality of English Language

Native English review suggested

Response: We have had the manuscript professionally edited by a native English-speaking editor through Wordvice (attached file).

Round 2

Reviewer 1 Report

Comments and Suggestions for Authors

Please review the following points:

  • Lines 86–87: Additional references are required.

  • Page 2: SGLT2 inhibitors should be mentioned in accordance with current diabetes guidelines.

Author Response

Responses to the comments of Reviewer #1

Comments and Suggestions for Authors

Please review the following points:

Lines 86–87: Additional references are required.

Response: Thank you very much for reviewing my manuscript again and for providing such valuable suggestions. We have added an additional reference in the description at Lines 86–87 (pages 2 and 14).

Page 2: SGLT2 inhibitors should be mentioned in accordance with current diabetes guidelines.

Response: We sincerely appreciate the reviewer’s valuable suggestion. In response, we have revised the manuscript to include a statement on SGLT2 inhibitors, in alignment with the current diabetes guidelines (page 2).

Reviewer 2 Report

Comments and Suggestions for Authors

Thank you for the clarifications. Unfortunately, this also highlights a more fundamental concern regarding the results of the paper. There appears to be a significant difference between the two groups with respect to the use of SGLT2 inhibitors. Given the well-established evidence that SGLT2 inhibitors alone substantially reduce both cardiovascular and microvascular events, it is plausible that the observed benefits may largely reflect the effects of SGLT2 inhibition, rather than a genuine difference between DPP-4 inhibitors and GLP-1 receptor agonists. To strengthen the validity of the conclusions, it would be highly informative if the outcomes could be presented separately for patients who did not receive SGLT2 inhibitors.

Author Response

Responses to the comments of Reviewer #2

Thank you for the clarifications. Unfortunately, this also highlights a more fundamental concern regarding the results of the paper. There appears to be a significant difference between the two groups with respect to the use of SGLT2 inhibitors. Given the well-established evidence that SGLT2 inhibitors alone substantially reduce both cardiovascular and microvascular events, it is plausible that the observed benefits may largely reflect the effects of SGLT2 inhibition, rather than a genuine difference between DPP-4 inhibitors and GLP-1 receptor agonists. To strengthen the validity of the conclusions, it would be highly informative if the outcomes could be presented separately for patients who did not receive SGLT2 inhibitors.

Response: Thank you very much for the valuable suggestion. We have additionally performed an outcome analysis excluding patients who received SGLT2 inhibitors in order to strengthen the validity of our conclusions. The sensitivity analysis yielded consistent results(pages 5 and 12, and Supplementary Table S3).

Reviewer 3 Report

Comments and Suggestions for Authors

I'm sorry, but the revisions made are of low quality and do not allow for a meaningful international interpretation in terms of scientific dissemination, especially for such a reputable journal. There is a complete lack of a real connection to clinical practice in relation to the data collected, which the authors justify inadequately and even without linking to other data presented (see lines 287–295). The suggested reporting tool (although correctly applied) lacks the appropriate reference citation—this represents a serious oversight in terms of intellectual property, with no credit given to the original authors. Moreover, the recommendations regarding limitations and “Perspectives for clinical practice” were disregarded, despite being addressed in the authors’ reply; this also indicates a lack of attention. A thorough review by a native English speaker is also strongly recommended.

Comments on the Quality of English Language

Native eglish review recommended

Author Response

Responses to the comments of Reviewer #3

I'm sorry, but the revisions made are of low quality and do not allow for a meaningful international interpretation in terms of scientific dissemination, especially for such a reputable journal. There is a complete lack of a real connection to clinical practice in relation to the data collected, which the authors justify inadequately and even without linking to other data presented (see lines 287–295). The suggested reporting tool (although correctly applied) lacks the appropriate reference citation—this represents a serious oversight in terms of intellectual property, with no credit given to the original authors. Moreover, the recommendations regarding limitations and “Perspectives for clinical practice” were disregarded, despite being addressed in the authors’ reply; this also indicates a lack of attention. A thorough review by a native English speaker is also strongly recommended.

Response: We are truly grateful for the reviewer’s constructive feedback, which has greatly contributed to improving the quality of our manuscript. In response, we have made the following revisions:

  1. The reference citation for the reporting tool has been added (pages 10 and 15).
  2. The discussion section has been expanded to include Perspectives for clinical practice and the study’s limitations(pages 9 and 10).
  3. The entire manuscript has been carefully reviewed, revised, and polished by a native English-speaking editor (see attached file).

Round 3

Reviewer 2 Report

Comments and Suggestions for Authors

I accept the answers.

Author Response

Responses to the comments of Reviewer #2

I accept the answers.

Response: We sincerely thank you for taking the time to review our manuscript, provide thoughtful feedback, and offer us such encouraging support.

Reviewer 3 Report

Comments and Suggestions for Authors

I'm sorry, but there is still a conflict regarding the transparency reporting checklist, which I cannot find among the supplementary files. I stand by my opinion, but I leave the final decision to the editor.

Author Response

Responses to the comments of Reviewer #3

I'm sorry, but there is still a conflict regarding the transparency reporting checklist, which I cannot find among the supplementary files. I stand by my opinion, but I leave the final decision to the editor.

Response: Thank you for your suggestion. We have added the reporting checklist as supplementary material (Supplementary File S1 — STROBE Checklist) to make the reporting of this study more complete. We appreciate your guidance.
